# AEmiGAP: AutoEncoder-Based miRNA–Gene Association Prediction Using Deep Learning Method

**DOI:** 10.3390/ijms252313075

**Published:** 2024-12-05

**Authors:** Seungwon Yoon, Hyewon Yoon, Jaeeun Cho, Kyuchul Lee

**Affiliations:** Department of Computer Science & Engineering, Chungnam National University, 99 Daehak-ro, Yuseong-gu, Daejeon 305-764, Republic of Korea; yoonenoch11@gmail.com (S.Y.); hyewony1001@gmail.com (H.Y.); jcho971028@gmail.com (J.C.)

**Keywords:** miRNA–gene association, deep learning, autoencoders, LSTM, cancer genomics, precision medicine, feature extraction, bioinformatics

## Abstract

MicroRNAs (miRNAs) play a crucial role in gene regulation and are strongly linked to various diseases, including cancer. This study presents AEmiGAP, an advanced deep learning model that integrates autoencoders with long short-term memory (LSTM) networks to predict miRNA–gene associations. By enhancing feature extraction through autoencoders, AEmiGAP captures intricate, latent relationships between miRNAs and genes with unprecedented accuracy, outperforming all existing models in miRNA–gene association prediction. A thoroughly curated dataset of positive and negative miRNA–gene pairs was generated using distance-based filtering methods, significantly improving the model’s AUC and overall predictive accuracy. Additionally, this study proposes two case studies to highlight AEmiGAP’s application: first, a top 30 list of miRNA–gene pairs with the highest predicted association scores among previously unknown pairs, and second, a list of the top 10 miRNAs strongly associated with each of five key oncogenes. These findings establish AEmiGAP as a new benchmark in miRNA–gene association prediction, with considerable potential to advance both cancer research and precision medicine.

## 1. Introduction

### 1.1. The Role of miRNAs in Gene Regulation and Cancer Development

MicroRNAs (miRNAs) are small, non-coding RNA molecules (18–25 nucleotides) that regulate gene expression post-transcriptionally, mainly by targeting the 3′ untranslated regions (3′ UTR) of specific messenger RNAs (mRNAs). This interaction allows miRNAs to either degrade mRNA or inhibit its translation, leading to precise control over protein synthesis [1]. miRNAs play critical roles in various biological processes, including cell differentiation, proliferation, apoptosis, and immune response. The dysregulation of miRNAs is associated with numerous diseases, particularly cancer [2], where abnormal miRNA levels can promote tumor growth, metastasis, and treatment resistance by acting as either oncogenes [3] or tumor suppressors.

In cancer, miRNAs influence essential traits like unchecked cell growth, resistance to cell death, angiogenesis, metastasis, and chemotherapy resistance. They function as potential biomarkers for early diagnosis and treatment monitoring due to their distinct expression profiles in healthy and malignant tissues. Furthermore, miRNAs found in body fluids, such as blood and urine, serve as non-invasive biomarkers for early cancer detection and therapeutic response tracking.

Understanding miRNA–gene interactions is essential for cancer genomics [4], as these associations reveal underlying molecular mechanisms and potential therapeutic targets. However, accurately identifying these interactions remains challenging due to the complexity of gene regulation networks and the diversity within cancer types. While experimental methods like reporter gene assays and high-throughput sequencing yield precise results, they are time-consuming and costly, making them unsuitable for large-scale studies. Recently, computational techniques, particularly machine learning and deep learning, have gained significant attention in the prediction of miRNA–gene interactions.

Deep learning models, especially, offer advanced tools for capturing the intricate, non-linear relationships between miRNAs and genes. These models excel in analyzing high-dimensional biological datasets and identifying patterns beyond the capabilities of conventional methods. By leveraging diverse datasets, deep learning approaches not only enhance prediction accuracy but also contribute to a deeper understanding of biological processes, paving the way for precision medicine [5]. The integration of deep learning in miRNA research enables researchers to handle the complexities of genomic networks with improved scalability and precision, ultimately contributing to tailored treatment strategies and better patient outcomes.

### 1.2. Related Works

To identify methodologies and limitations in deep learning-based studies focused on miRNA associations, we compiled research centered on the association prediction of miRNAs with all possible associated non-coding RNAs (mRNAs, lncRNAs, circRNAs) and genes, as summarized in Table 1. This table presents a compilation of deep learning-based studies organized by our research team, offering insights into the diverse applications of deep learning techniques in genomics. We organized the related works from the perspectives of datasets, methods for generating negative data, embedding techniques, and the types of deep learning models applied, highlighting the innovative methods utilized in previous studies.

As demonstrated, deep learning models for predicting miRNA associations have been actively developed. Most of these studies share common characteristics: they construct training datasets using well-known open data sources and label association information accordingly. Additionally, negative sets are commonly created through random sampling, various vectorization methods are applied for embedding, and sequence-specialized deep learning models are employed for classification. However, there is still room for improvement in these approaches.

To enhance performance, larger training datasets must be developed to allow deep learning models to learn from a broader spectrum of interactions. Furthermore, negative sets must be produced using more sophisticated techniques instead of random sampling. Utilizing embedding approaches specifically designed for genetic data might enhance model accuracy. Ultimately, there is a need for even higher-performing deep learning models in association prediction. Recent improvements indicate that miRNA association prediction models have significant potential for further enhancement. Our research aims to overcome the limitations of prior studies by implementing a deep learning model that achieves superior performance in miRNA–gene association prediction, therefore setting a novel standard in predictive accuracy and robustness.

### 1.3. Overview of the AEmiGAP Workflow

In a prior study, we introduced miGAP [51], a deep learning model based on long short-term memory (LSTM) networks, designed to predict miRNA–gene associations. Among all related studies listed in Table 1, miGAP is distinguished by its use of the most extensive dataset, the construction of a highly refined negative set, and an innovative embedding method, achieving top-tier performance in miRNA–gene association prediction. However, there remains room for improvement in miGAP to further enhance its predictive capabilities. Specifically, we aim to improve classification performance through more sophisticated feature extraction techniques, allowing the model to better capture the complex biological patterns inherent in miRNA and gene expression data.

We suggest the implementation of AEmiGAP, a sophisticated framework that combines autoencoders (AEs) [52] with the miGAP architecture. Autoencoders, a type of unsupervised deep learning, are highly suitable for acquiring compact representations of input data while maintaining crucial information. Our goal is to boost the performance of miRNA–gene association prediction by integrating AEs into the current miGAP framework. This integration will improve the model’s capability to extract concealed characteristics from intricate biological datasets. AEmiGAP specifically aims to achieve high-performance miRNA–gene association prediction by enhancing the deep learning model’s ability to capture genomic features more effectively, ultimately uncovering previously unknown miRNA–gene associations through an advanced deep learning framework.

The workflow of AEmiGAP consists of several key stages that work together to predict miRNA–gene associations, as illustrated in Figure 1. The following steps outline the process:(1)**[Embedding]**—*miRNA and Gene Data Embedding*: A miRNA set is obtained from the miRBase database, and a gene set is collected from BioMart. Both sets are encoded into vectors using Protein2Vector to ensure that biological features are preserved. This representation allows the model to capture meaningful patterns between the miRNA and gene sequences.(2)**[Labeling]**—*Negative Data Filtering*: Negative data are filtered based on distance metrics such as the Euclidean, cosine, and Mahalanobis distances [53]. This step ensures that the negative examples used during training are sufficiently distinct, improving the quality of the model’s predictions by minimizing noise.(3)**[Feature Extraction]**—*Autoencoder for Latent Feature Extraction*: An autoencoder takes the miRNA–gene input pairs and extracts compact, meaningful latent vectors. These vectors capture the most important underlying patterns from the miRNA and gene sequences, helping to enhance the model’s ability to make accurate predictions. The latent vectors are added back to the original data.(4)**[Association Prediction]**—*LSTM-Based Classification*: The enriched data, containing miRNAs, genes, latent vectors, and labels, are input into an LSTM network. The LSTM is capable of capturing sequential dependencies and complex relationships within the data, ultimately predicting miRNA–gene associations with high precision.

This workflow allows the AEmiGAP model to integrate both biological sequence data and latent feature representations, which significantly improves its ability to predict meaningful miRNA–gene associations.

This paper introduces AEmiGAP as a novel approach and an extensive structure for predicting miRNA–gene associations. AEmiGAP integrates autoencoders for feature extraction and utilizes LSTM-based architectures, achieving superior predictive accuracy while providing a scalable and adaptable methodology for genomic research. This framework aims to serve as a comprehensive instrument for researchers, promoting the investigation of miRNA–gene interactions and supporting the experimental verification of new correlations.

The structure of this paper is organized as follows: Section 2 presents the experimental results, including the performance evaluation of the model and case studies highlighting its application. Section 3 provides a comprehensive discussion of the findings and their implications. Section 4 details the dataset and methodologies used to construct and evaluate AEmiGAP. Finally, Section 5 concludes with a summary of the key contributions, limitations, and future research directions.

## 2. Results

### 2.1. Dataset Description

Our research work utilizes a dataset containing a collection of miRNA–gene correlations to effectively conduct deep learning studies. The dataset, developed by our research team, consists of 2656 miRNA sequences and 14,319 gene sequences, resulting in a total of 717,728 miRNA–gene combinations. This dataset is well balanced, with 358,864 positive and 358,864 negative samples purposefully included to ensure balanced and robust learning, thereby enhancing the reliability of the classification model’s performance. To predict miRNA–gene associations, we constructed this comprehensive dataset by integrating experimentally validated interaction pairs from miRTarBase with sequence information from miRBase (for miRNA sequences) and biomaRt (for gene sequences). The dataset encompasses a diverse range of human genes and their corresponding sequences, ensuring robustness and comprehensiveness for predictive modeling.

To optimize the dataset for deep learning applications, both miRNA and gene sequences were vectorized using the Protein2Vec embedding method [54]. This technique captures the key sequential and structural features of the sequences, providing a compact and informative representation for the precise exploration of miRNA–gene interactions. Due to its size, quality, and preprocessing, this dataset is ideal for exploring the interactions between miRNA and genes with precision and applicability.

### 2.2. Autoencoder-Based miRNA–Gene Association Prediction Deep Learning Model Performance

#### 2.2.1. Autoencoder in Extracting miRNA–Gene Features

An autoencoder was employed to enhance the classification accuracy of our learning model. An autoencoder is an unsupervised network designed to generate representations of input data (encoded information) primarily for dimensionality reduction. It comprises two components: an encoder, which compresses the input into a latent space representation, and a decoder, which reconstructs the input from this compressed format. Figure 2 illustrates the construction of three encoder layers and three decoder levels. Each layer incrementally captures increasingly intricate aspects of the miRNA and gene sequences. The latent vector, positioned between the encoder and decoder, incorporates the most critical information from the sequences. This approach enables the model to acquire the complex patterns essential for enhancing miRNA–gene connection prediction. Through the meticulous adjustment of each layer, the network minimizes redundant information, thereby retaining the critical traits necessary for precise classification.

After completing the autoencoder training, we evaluated its effectiveness. Both the training and testing errors converged after around 20 epochs, confirming that the model successfully learned to compress and reconstruct the data without a significant loss of information. This confirms that the feature extraction process enhanced our model’s ability to efficiently represent miRNA–gene interactions, ultimately improving its classification performance.

Our model’s autoencoder was designed to effectively capture linear features from miRNA and gene sequences. This approach was crucial due to the limitations of feature extraction methods in hidden connections and intricate patterns in biological data, particularly in highly variable sequences, such as miRNAs and genes. We completed feature extraction in two steps using the autoencoder technique. Initially, the miRNA and gene sequences were transformed into vectors via Protein2Vec, which encapsulated their structural characteristics. We employed an autoencoder to enhance the representations of the interactions between the miRNA and genes. This enabled the deep learning model to concentrate on the pertinent and unique sequence characteristics.

The encoder component of the autoencoder compresses the input data into a representation by eliminating superfluous information and noise while preserving essential sequential properties. The decoder subsequently rebuilds the input to guarantee that no essential information was lost during compression. After the initial autoencoder process was completed, we conducted a series of experiments to investigate how different approaches to utilizing the vectors could improve the prediction of miRNA–gene relationships.

#### 2.2.2. Evaluation of Latent Vector Integration for Enhanced miRNA–Gene Interaction Prediction

We initially performed an experiment in which the condensed features produced by the trained autoencoder were directly input into the deep learning classification model (Experiment A). This enabled us to assess the efficacy of the autoencoder’s latent vectors, which encapsulated the condensed representations of the miRNA and gene sequences, in predicting miRNA–gene relationships. The objective was to evaluate the predictive capability of the autoencoder’s output only. In a second experiment, we integrated these latent vectors with the original sequence data using Protein2Vec to enhance prediction accuracy.

In the second experiment, we carried out another test by combining the vectors with the initial vectorized sequence information from Protein2Vec and using them together as inputs for the classification model (Experiment B). This allowed us to assess whether blending the autoencoder’s summarized characteristics with the original sequence data could enhance our ability to predict outcomes.

Experiments A and B were conducted to determine how the autoencoder could be effectively used to improve the prediction of miRNA–gene interactions in a study where the complete dataset was split into training and testing sets at a ratio of 8 to 2. We employed the LSTM-based miGAP model within a learning framework known for its effectiveness in predicting connections between miRNAs and genes. To evaluate how different dimensionalities affect the models capabilities, we tested various latent vector sizes, such as 8, 16, 32, 64, and 128. The experiments aimed to determine how various latent vector sizes and integration methods affect the model’s performance.

We used the Area Under the Curve (AUC) as the measure to fully evaluate how well the models performed in Experiments A and B. The AUC is a respected metric in RNA research in tasks such as predicting miRNA–gene relationships. Our goal was to assess how effective the enhanced models were at distinguishing between inaccurate predictions by comparing the AUC values with different latent vector sizes and integration methods. We were able to reliably assess the models’ classification skills by using the AUC, which considers both sensitivity and specificity to ensure that the model performs well with relationships and can also effectively adapt to new datasets. Furthermore, the AUC metric allowed us to gauge improvements in performance as the latent vector sizes increased, providing insights into the configurations for accurately predicting associations between miRNA and genes. The results of the two experiments are summarized in Table 2.

The results of the experiment in Table 2 show that using the vectors produced by the autoencoder (Experiment A) improved performance as the size of the hidden vector increased, reaching an AUC of 0.9331 at 128 dimensions. However, the best performance was observed in Experiment B, where the hidden vectors were combined with the sequence data from Protein2Vec. This method consistently outperformed Experiment A in all vector dimensions and reached an AUC of 0.9846 with a vector size of 64; this suggests that combining both representations enhances predictive accuracy. The results of this study demonstrate a method for improving model efficiency using latent vectors derived from autoencoders. By integrating these vectors, which encapsulate condensed features from miRNA and gene sequences, with the primary sequence data derived from protein embeddings (Protein2Vec), we consistently achieved strong results. This approach yielded a high AUC score, particularly with a latent vector size of 64. Our research findings show that the best method to improve classification accuracy is to combine the data with their condensed latent representations to provide an input for the deep learning classification model.

#### 2.2.3. Performance Analysis of Autoencoder-Based Feature Extraction for miRNA–Gene Associations

This section comprehensively examines the efficacy of well-executed feature extraction in predicting miRNA–gene associations. Previous assessments indicate that integrating latent features derived from autoencoders into classification learning models markedly enhances predictive accuracy. To comprehensively assess the effectiveness of the proposed method, we employed a 5-fold cross-validation technique, which is very beneficial in deep learning applications utilizing biological datasets. This methodology offers numerous advantages: it guarantees thorough testing of the model across various data subsets, reducing the likelihood of overfitting and delivering an extensive evaluation of the model’s generalizability. By dividing the dataset into five equal segments, 5-fold cross-validation allows each segment to serve as a test set once, while the remaining four segments are used for training. This process guarantees that the model’s performance is evaluated across diverse portions of the dataset, thereby reducing potential biases and providing a more reliable measure of predictive accuracy.

The mean performance measures, including precision, recall, accuracy, F1 score, and AUC, were calculated across all five folds to evaluate the model’s efficacy. Table 3 indicates that the model achieved outcomes across all instances with the following metrics: a precision of 96.58%, a recall of 92%, an accuracy of 94%, an F1 score of 95%, and an AUC of 98%. The findings illustrate the efficacy of the suggested method in detecting miRNA–gene connections while reducing false predictions.

In these tests, we employed a learning model founded on the LSTM architecture. This method has demonstrated significant efficacy in predicting relationships between miRNA and genes. To further boost its performance, we merged the miRNA and gene sequence feature representations produced by the autoencoders alongside the actual sequence data added via pro-tein2Vec. The model effectively used both the raw sequence data and the summary characteristics through this integration, enhancing its comprehension of the complex interactions between miRNAs and genes.

Table 3 shows that, among the 5-fold cross-validation findings, Fold-4 demonstrates superior performance based on a comparison of the Receiver Operating Characteristic (ROC) curve and confusion matrix charts. The AUC value for the ROC curve of Fold-4 is 0.9857 (Figure 3), a demonstration of its exemplary efficacy in precisely differentiating between positive and negative interactions with miRNA–gene pairs. As illustrated in Figure 3, the confusion matrix indicates that the model demonstrates excellent accuracy, properly identifying 66,099 true positives and 70,048 true negatives while exhibiting minimal false positives (1723) and false negatives (5618).

The AUC metric is commonly employed in RNA research to predict correlations between miRNA and genes, assessing the balance between sensitivity and specificity across all thresholds. This thorough evaluation of efficacy renders it an excellent option for assessing models in the domain of biology. Utilizing the AUC provides an accurate representation of the model’s classification performance by successfully differentiating between correct and wrong predictions. This statistic facilitates the comparison of models in research projects and datasets.

Moreover, our results indicate that the optimal method for enhancing the precision of miRNA–gene association classification involves integrating raw sequencing data with compressed representations produced by autoencoders. This amalgamation of methodologies enables the model to use both characteristics and critical biological indicators, resulting in enhanced accuracy. The application of autoencoders for feature extraction demonstrates efficacy, and we foresee its significance in propelling future research in this domain.

### 2.3. Case Study for the Discovery of Novel Associations

#### 2.3.1. Construction of the Case Study Dataset and Methodology

Our study utilized data focused on connections, such as interactions between miRNAs and genes that have been widely used and validated through the miRTarBase database—recognized as a resource for experimentally proven microRNA (miRNA) targets—making it a reliable foundation for creating a trustworthy set of established miRNA–gene associations in our analysis. Henceforth, the positive dataset incorporated in this study is considered representative of the majority of confirmed miRNA–gene interactions.

As shown in Table 4, we developed an Unknown Data Pool to explore the connections between miRNAs and genes in our research project by cross-referencing all 2656 miRNAs and 14,319 genes to create a comprehensive pool of possible miRNA–gene pairs. To ensure that the data remained fresh and untainted by known associations, we excluded any interactions between miRNAs and genes already documented in miRTarBase. This process allowed us to focus solely on discovering novel miRNA–gene combinations of established links. Through this approach, we successfully established a dataset allowing for the uncovering of yet-undiscovered connections between miRNAs and genes.

Our case study aimed to predict miRNA–gene interactions that could have a correlation but were previously unknown to us. To achieve this goal, we employed the AEmiGAP model within a learning framework that had shown good performance in previous evaluations. For our study, we used the model trained on Fold-4, as it had the highest Area Under the Curve (AUC) score during the 5-fold cross-validation process.

In the case study discussed in this research paper, we combined the characteristics generated by the autoencoder with the miRNA and gene sequence information. This combined dataset was then input into the learning model. Subsequently, the algorithm predicted the degree of correlation for each miRNA–gene pair in the dataset. Based on the predicted scores, the miRNA–gene pairs were arranged in descending order, ranked according to how likely they are to be associated. Among the ranked results, the top 30 miRNA–gene pairs from the entire Unknown Data Pool are presented. Additionally, we highlight the top 10 miRNA–gene associations most closely related to five well-known oncogenes. As suggested by the model’s high scores, these pairs represent potentially novel associations, holding significant promise for biological validation in future research efforts.

Our research aimed to discover connections between miRNA and genes by using the prediction abilities of the AEmiGAP model, which combines features generated by the autoencoder with the original sequence data to focus on unexplored miRNA–gene pairs, as part of our methodology for enriching the current knowledge on miRNA–gene interactions and suggesting new avenues for future research endeavors.

#### 2.3.2. Top 30 Predicted miRNA–Gene Associations from the Unknown Data Pool by AEmiGAP

This section of our study report presents the 30 combinations of miRNA–gene pairs found by our learning system in the Unknown Data Pool, exhibiting the highest probability of correlation. These associations are absent from the miRTarBase database and may suggest novel connections that remain unverified by experimental evidence. To evaluate their significance in a biological context, these miRNA–gene pairs were compared with data from the TargetScan [55] and miRWalk [56] databases, both of which include information from validated miRNA–gene interactions derived from experimental evidence.

The top 30 miRNA–gene pairings, ranked according to the prediction scores of the AEmiGAP model, are shown in Table 5. The most consistently predicted miRNAs in this table are hsa-miR-92a-3p and hsa-miR-548ah-5p. This suggests important roles for these miRNAs in hitherto unexamined gene regulation networks. Notably, hsa-miR-92a-3p emerges as the top-ranked miRNA in several predictions, suggesting possible undiscovered biological functions associated with gene regulation. The repeated correlation of hsa-miR-548ah-5p with various genes also necessitates further research.

Several genes of considerable intrigue, such as LRRFIP1 and PTPN12, are repeatedly associated with transcription regulation and cancer-related pathways. This is corroborated by data from TargetScan and miRWalk. The C1D gene exhibits potential due to its predicted association with hsa-miR-92a-3p, particularly in relation to the DNA damage response, rendering it a noteworthy target for biological research validation endeavors.

The amalgamation of these top-ranked miRNA–gene pair findings yields a list for subsequent experimental validation, which could help elucidate the mechanisms by which miRNAs regulate gene activity in novel research domains. The data in Table 5 indicate that certain miRNA–gene pairings are identified as potential associations by TargetScan or miRWalk but have not yet been incorporated into miRTarBase, thus reinforcing the credibility of the model’s predictions.

This case study demonstrates the efficacy of using characteristics obtained from autoencoders alongside miRNA and gene sequencing data to provide new biological insights.

#### 2.3.3. Prediction of miRNAs with High Association with Prominent Oncogenes Using the AEmiGAP

Predicting miRNA–gene associations is critically important in cancer research, as these interactions can reveal mechanisms of tumor growth, progression, and resistance to treatment. miRNAs, acting as either oncogenes or tumor suppressors, regulate pathways that control cell survival, proliferation, and metastasis. Understanding these interactions can inform targeted therapies, provide biomarkers for early detection, and offer insights into personalized cancer treatments.

For this case study, we focused on identifying miRNA–gene interactions associated with prominent oncogenes to uncover novel insights into cancer biology [57]. To select the most impactful oncogenes, we reviewed gene lists from four major sources:MSK IMPACT panel [58], which analyzed 505 genes for genetic alterations across diverse cancers.Foundation One CDx panel [59], which included 324 genes relevant to cancer diagnosis and therapies.Vogelstein et al.’s cancer genomics study [60], which identified 125 widely recognized oncogenes.COSMIC Cancer Gene Census Tier 1 database [61], which catalogs 581 genes critical to cancer research.

By cross-referencing these sources, we identified five top oncogenes—ALK, EGFR, GNAS, KRAS, and PIK3CA [62]—due to their frequent occurrence and their critical roles in multiple cancers. These genes are strongly implicated in tumor growth, metastasis, and treatment resistance. Importantly, their sequences were already included in our dataset, enabling the use of AEmiGAP for predictive analysis.

ALK (anaplastic lymphoma kinase) mutations are commonly found in non-small-cell lung cancer (NSCLC). These mutations lead to the abnormal activation of the ALK protein, which promotes cancer cell survival and proliferation, making ALK inhibitors an effective treatment in targeted cancer therapies.EGFR (epidermal growth factor receptor) is a gene frequently mutated in NSCLC. These mutations result in the overactivation of the EGFR protein, causing uncontrolled cell growth.GNAS (guanine nucleotide binding protein, alpha stimulating) mutations are observed in bone marrow and pancreatic cancers. These mutations disrupt signaling pathways, leading to abnormal cancer cell growth and tumor formation.KRAS (Kirsten rat sarcoma virus) mutations are commonly found in colorectal, pancreatic, and non-small-cell lung cancers. These mutations activate signaling pathways that promote cancer cell growth, playing a critical role in tumor progression and metastasis.PIK3CA (phosphatidylinositol-4,5-bisphosphate 3-kinase catalytic subunit alpha) mutations are commonly seen in breast and cervical cancers. These mutations regulate cancer cell survival and growth by activating the PI3K/AKT signaling pathway, which can contribute to resistance against cancer treatments.

Using AEmiGAP, we predicted miRNA associations with these five oncogenes that are not currently reported in miRTarBase, one of the most comprehensive databases for experimentally validated miRNA–gene interactions. Table 6, Table 7, Table 8, Table 9 and Table 10 indicate the top 10 miRNAs with the most relevant predicted relationships for each oncogene. These findings emphasize AEmiGAP’s capacity to identify novel miRNAs associated with oncogenes, enhancing our comprehension of cancer growth and informing the creation of targeted therapeutics and precision medicine strategies in cancer research.

The ALK gene (Table 6) is notable because hsa-miR-6081, hsa-miR-4708-3p, and hsa-miR-1225-3p are frequently associated with it. These miRNAs have not been previously documented in miRTarBase but have been recognized as possible targets by miRWALK. Considering the role of ALK in non-small-cell lung cancer, our findings suggest potential pathways for further exploration of the miRNA regulation of ALK in cancer progression.The GNAS gene (Table 7) is predicted to have a strong association with hsa-miR-519c-5p, hsa-miR-4485-3p, and hsa-miR-518d-5p. Similar to ALK, these miRNAs are absent from miRTarBase but present in miRWALK, indicating potential new regulatory mechanisms that may affect GNAS expression. GNAS is frequently associated with several malignancies, such as bone marrow and pancreatic cancers, making these predictions crucial for further investigation.The KRAS gene (Table 8) shows significant correlations with hsa-miR-25-3p, hsa-miR-4753-3p, and hsa-miR-6826-3p. Notably, several miRNAs (hsa-miR-25-3p and hsa-miR-4753-3p) are supported by TargetScan, suggesting that these predictions have a higher likelihood of biological significance. Given the critical role of KRAS mutations in cancers such as colorectal and pancreatic tumors, these findings suggest possible new targets for therapeutic intervention.The EGFR gene (Table 9) is anticipated to be regulated by several miRNAs, including hsa-miR-25-3p, hsa-miR-4685-3p, and hsa-miR-6794-3p. Many of these miRNAs are corroborated by both miRWALK and TargetScan, thereby strengthening the credibility of these predictions. Since EGFR mutations are common in non-small-cell lung cancer and are associated with cancer cell proliferation, these miRNA–gene correlations may be crucial for developing novel cancer therapies.Regarding the PIK3CA gene (Table 10), hsa-miR-25-3p, hsa-miR-6826-3p, and hsa-miR-6780a-3p are identified as the top candidates for its regulation. A significant finding is the recurrent presence of hsa-miR-25-3p across multiple genes, suggesting its potential broad regulatory role in cancer pathways. The strong support from both miRWALK and TargetScan for some miRNAs linked to PIK3CA emphasizes their importance in cancer biology, particularly in breast and cervical cancers, where PIK3CA mutations are prevalent.

The predictive studies concerning these five notable oncogenes and their corresponding miRNAs reveal the AEmiGAP model’s proficiency in detecting unique and possibly significant miRNA–gene correlations. This provides significant insights that may inform future research on cancer biology and miRNA regulation. By identifying these miRNA–oncogene associations, AEmiGAP opens new possibilities for early diagnosis, treatment personalization, and targeted therapy development in oncology. Recurrent miRNAs, including hsa-miR-25-3p, hsa-miR-6081, and hsa-miR-4708-3p, are observed in several gene relationships, indicating their potential significance in the regulation of critical cancer-related pathways. These miRNAs may represent viable candidates for additional experimental validation, underscoring their potential involvement in cancer regulation.

The robust backing from databases such as miRWALK and TargetScan for several anticipated miRNA–gene combinations bolsters the reliability of the model’s predictions. This underscores the potential of these miRNA–oncogene interactions to serve as biomarkers or therapeutic targets, offering fresh insights into cancer genesis and progression. As a result, AEmiGAP’s predictions provide a basis for further research aimed at uncovering novel cancer treatments and improving patient outcomes in oncology.

## 3. Discussion

### 3.1. Comparison with Other State-of-the-Art Studies

#### 3.1.1. Distinctions Between AEmiGAP and miGAP

The AEmiGAP model is an enhancement of the miGAP framework, originally created by our research team to predict miRNA–gene associations. Although miGAP exhibits a robust predictive capability in discerning miRNA–gene associations, AEmiGAP markedly enhances its predecessor by incorporating novel methodologies for feature extraction, sophisticated data management, and improved learning models, all of which augment its performance.

The primary distinction between AEmiGAP and miGAP is observed in feature extraction. miGAP utilizes conventional feature representations, whereas AEmiGAP includes autoencoder-based feature extraction. AEmiGAP introduces an innovative approach by concatenating the additional feature vectors extracted from the autoencoder with the original data, effectively applying autoencoders in the field of RNA association prediction. By enabling the deep learning model to learn not only from the sequence vectors of miRNAs and genes but also from their extracted features, AEmiGAP achieves improved association predictive accuracy. This method enhances the model’s ability to generalize and identify intricate, non-linear patterns in the data, resulting in increased predictive accuracy.

Alongside enhanced feature extraction, comparative analyses of model performance between AEmiGAP and miGAP demonstrate significant advancements in critical assessment measures. Table 11 presents a detailed summary of these performance improvements. AEmiGAP showed improvements in precision, recall, accuracy, the F1 score, and the AUC. Precision rose by 2.97%, recall by 0.06%, accuracy by 1.24%, F1 score by 1.50%, and the AUC by 0.43% on average. The AUC (Area Under the Curve) is a crucial measure in miRNA–gene association research as it assesses a model’s capacity to differentiate between positive and negative relationships. An elevated AUC value indicates that the model has an enhanced capacity to accurately categorize miRNA–gene interactions, a critical aspect of biological research.

An elevated AUC, along with improvements in precision, recall, accuracy, and F1 score, indicates that AEmiGAP has an enhanced ability to accurately categorize miRNA–gene interactions, a critical aspect of biological research. These performance metrics collectively demonstrate that AEmiGAP effectively captures and distinguishes the complex characteristics of miRNAs and genes, resulting in more precise and reliable predictions. A model with such comprehensive performance gains is better equipped to identify true miRNA–gene interactions with minimal false positives and negatives, which is essential for advancing accurate insights and applications in genomics research.

AEmiGAP expands upon the success of our initial miGAP model, introducing crucial improvements that greatly enhance prediction accuracy and the ability to discover new associations. AEmiGAP surpasses miGAP in all key performance metrics by utilizing improved deep learning algorithms and enhanced feature extraction methods, as shown in Table 11.

#### 3.1.2. Comparison with Other Related Studies

AEmiGAP stands out among recent studies on miRNA–gene association prediction and all miRNA association prediction research, exhibiting exceptional AUC performance. Table 12 presents a comparative analysis of AEmiGAP’s performance relative to other models, highlighting its superior standing across key metrics. AEmiGAP adeptly integrates miRNA and gene embeddings from Protein2Vec with autoencoder-driven feature extraction and LSTM-based learning to effectively capture both linear and non-linear relationships in sequence data. This multi-step methodology achieves an AUC of 0.9857, markedly surpassing earlier models such as SG-LSTM FRAME (AUC: 0.9363) and SRG-Vote (AUC: 0.95). These findings underscore AEmiGAP’s reliability and precision in identifying miRNA–gene associations.

AEmiGAP’s robust performance can be attributed to both its sophisticated architecture and the research team’s meticulously crafted data preparation process. Table 12 demonstrates that AEmiGAP comprises 358,864 positive and negative miRNA–gene associations, markedly exceeding previous models. Instead of merely relying on bigger data quantities, our team carefully processed the dataset by extracting, vectorizing, and labeling each miRNA–gene relationship, thereby ensuring the data’s high quality and biological worth. This methodology yielded a dataset markedly larger than those utilized in previous research and guaranteed enhanced data quality via an optimized embedding strategy specifically designed for miRNA and gene attributes, enabling the model to more effectively capture critical biological complexity compared to traditional methods. This carefully curated dataset underscores AEmiGAP’s scalability and its ability to handle complex datasets with precision, allowing it to capture nuanced patterns in biological sequences that might be missed in less rigorously constructed datasets.

AEmiGAP features a sophisticated negative data filtering method that sets it apart from previous models. The model employs Euclidean, cosine, and Mahalanobis distances to select negative pairs that markedly differ from positive pairs, thereby enhancing the precision of non-interaction representations. This approach reduces false negatives and improves classification accuracy, with Mahalanobis distance introducing an additional degree of complexity by accounting for data covariance, effectively mitigating outliers and preventing erroneous associations.

Through the integration of advanced embedding techniques and selectively refined negative datasets, AEmiGAP has significantly improved miRNA–gene association prediction. By capturing complex sequential patterns and maintaining data integrity through autoencoding, AEmiGAP demonstrates notable advancements in predictive accuracy and scalability. These enhancements are especially valuable for uncovering novel miRNA–gene connections, potentially deepening insights into disease mechanisms, particularly in cancer research.

## 4. Materials and Methods

### 4.1. Dataset

To create the AEmiGAP model, we utilized the most extensive dataset on miRNA–gene associations, which was previously introduced in our miGAP study. This dataset was used as the basis for building a strong predictive model that includes a diverse range of interactions between miRNA and genes. The dataset was constructed by amalgamating information from multiple prominent public sources, with each database providing essential data. We specifically employed miRTarBase to gather experimentally confirmed miRNA–gene interactions, miRBase to obtain miRNA sequence data, and biomaRt to gain gene sequence information. Both miRNA and gene sequences in this dataset are derived from human sources, ensuring relevance to human biology and comprehensive coverage for predictive modeling.

Initially, miRTarBase, the most comprehensive database for miRNA–gene associations, offered a total of 380,634 experimentally validated correlations between miRNAs and genes. During the refinement process, we considered only pairs where sequence information was available in both miRBase (for miRNAs) and biomaRt (for genes). This filtering resulted in a final dataset of 358,864 positive pairs, consisting of 2656 unique miRNAs and 14,319 unique genes.

To provide a well-balanced dataset, negative samples were generated using three distance-based criteria: the Euclidean distance, cosine similarity, and Mahalanobis distance. The criteria were used to pick miRNA–gene combinations that do not interact, which were then used as negative examples to train the model. Out of a pool of 4,932,554 potential negative samples, 358,864 miRNA–gene pairings that do not interact were chosen at random to equal the number of positive pairs. The dataset used for training and validating the AEmiGAP model consisted of both positive and negative miRNA–gene connections. This balanced dataset was crucial in establishing a strong foundation for the accurate prediction of miRNA–gene interactions.

### 4.2. Feature Embedding

#### 4.2.1. Sequence Vectorization

To enhance model performance, it is crucial to have an accurate representation of sequence data for predicting miRNA–gene correlations. miRNA sequences, composed of nucleotides (A, U, C, G), and gene sequences, composed of DNA bases (A, T, C, G), necessitate strong embedding techniques to capture crucial sequence characteristics that impact their interactions. Conventional techniques such as one-hot encoding are hindered by their high dimensionality and limited capacity to capture significant connections between sequence segments. To overcome this constraint, we initially implemented the Protein2Vec embedding method, which is an expansion of the well-employed Doc2Vec procedure, to transform both miRNA and gene sequences into compact vector representations. This approach effectively captures the sequential and structural attributes of both miRNAs and genes, allowing the model to comprehend their intricate regulatory functions. The sequences were converted into 64-dimensional vectors, resulting in a concise yet informative representation of the original sequence data. A comprehensive dataset for predicting miRNA–gene correlations was created by embedding a total of 2656 miRNA and 14,319 gene sequences.

#### 4.2.2. Feature Extraction with Autoencoder

To address the requirement of more advanced feature extraction, we expanded the embedding process by implementing an additional stage that included autoencoders (AEs). Although the original embedding using Protein2Vec successfully captured fundamental sequence characteristics, it was insufficient for capturing complex, non-linear connections between miRNAs and genes, and more sophisticated approaches were therefore necessary. Autoencoders, an unsupervised deep learning model, are highly effective at revealing latent, high-dimensional features that are not captured when using conventional embedding approaches.

Consequently, the feature extraction procedure was carried out in two separate steps, with each stage making a distinctive contribution to the model’s capacity to capture and interpret miRNA–gene interactions. During the initial phase, Protein2Vec was utilized to convert the raw sequence data into vectors with a dimensionality of 64. The initial embedding successfully caught the fundamental sequential and structural characteristics of the nucleotide sequences, which are crucial for comprehending the fundamental interactions between miRNAs and their target genes. Protein2Vec is a modified version of word embedding models, which are frequently used in natural language processing. It retains the positional context and sequence relationships, resulting in a valuable representation of biological sequences. Although this technique provides useful information about the overall characteristics of the sequences, it has limitations in detecting more complicated, non-linear patterns that are crucial for accurately predicting miRNA–gene interactions, especially in complex biological systems such as cancer.

To address these constraints and improve the model’s ability to represent data, we implemented a secondary feature extraction phase with autoencoders. Following the integration of the miRNA and gene sequences, we subjected these embeddings to additional processing using an autoencoder network, which is particularly effective in capturing more complex and underlying characteristics within the data. The autoencoder was assigned the objective of acquiring compressed representations of the embedded sequences, thereby enabling the model to extract latent properties that might not be captured when only using conventional embedding techniques.

The encoder component of the autoencoder compresses the original 128-dimensional embeddings (miRNA and gene) into a smaller, lower-dimensional latent space. This compression forces the network to extract the most significant features of the sequence data, eliminating irrelevant or redundant information. Mathematically, this process is described by the following equation:(1)$$z=f\left(x\right)=\sigma (Wx+b).$$
where x is the input (the original sequence embeddings), W represents the weights, b is the bias term, and σ is the activation function. The latent vector z captures the key features of the input data.

Subsequently, the decoder reconstructs the input data from the compressed latent representation z, aiming to retain the critical information throughout the feature extraction process. This step is defined by the following equation:(2)$$\hat{x}=g\left(z\right)=\overset{´}{\sigma }(W^{\prime }z+b^{\prime }).$$
where $W^{\prime }$ and $b^{\prime }$ are the weights and bias of the decoder, and $\hat{x}$ represents the reconstructed input. The autoencoder learns to preserve the essential qualities of the original embeddings by minimizing the reconstruction loss between x and $\hat{x}$.

Through this process, the autoencoder not only retains the most important features of the original sequence embeddings but also generates additional features that capture higher-order interactions between miRNAs and genes, thereby enhancing the model’s predictive capabilities.

The autoencoder architecture utilized in this work was configured with particular parameters to optimize feature extraction efficacy. The architecture had three fully connected layers for both the encoder and decoder, enabling the capture of hierarchical characteristics from the input data and the generation of strong latent representations. The latent space dimensionality was established at 64, signifying the condensed feature space that preserves the most essential characteristics of the input data.

In the encoder layers, the Rectified Linear Unit (ReLU) was utilized as the activation function, facilitating accelerated convergence and enhanced feature extraction through the introduction of non-linearity. The decoder layers utilized a sigmoid activation function to facilitate the precise reconstruction of the original input data. The Adam optimizer, selected for its adjustable learning rate and efficacy in gradient descent, was employed with a learning rate of 0.001, striking a balance between convergence speed and model stability. The training utilized a batch size of 128, facilitating efficient sample processing per iteration and optimizing computing efficiency. The employed loss function was the mean squared error (MSE), which facilitated the minimization of reconstruction loss between the original input and the reconstructed output. The model underwent training for 200 epochs, allowing a sufficient duration to efficiently learn the compressed representations while mitigating overfitting. The dataset was divided in a 9:1 ratio for training and testing, providing ample data for model training while preserving a segment for validation to assess generalization performance. The amalgamation of parameters and the training approach allowed the autoencoder to proficiently augment the feature extraction process and facilitate enhanced miRNA–gene association predictions in the AEmiGAP model.

The latent feature vectors obtained were subsequently combined with the original Protein2Vec embeddings, thereby enhancing the dataset with both the original sequence-level information and the newly derived latent features. The utilization of this dual-stage feature extraction strategy enables the deep learning model to process a more extensive and refined representation of the data, hence greatly enhancing its ability to identify minor miRNA–gene connections that could otherwise go unnoticed.

Overall, the utilization of autoencoders in this two-step procedure not only increases the complexity of the feature space but also enables the model to make more accurate predictions across different biological datasets. The use of this advanced method for extracting features is essential to accurately capture the intricacy of miRNA–gene regulation networks, especially in the context of illnesses such as cancer, where complicated and non-linear interactions frequently determine gene expression patterns. The subsequent sections of this manuscript include a comprehensive examination of the autoencoder’s intricate architecture and implementation details. Additionally, a thorough analysis of the experimental outcomes showcases the enhanced prediction accuracy attained through the utilization of this technology.

### 4.3. Negative Data Creation

In AEmiGAP, the creation of a strong and dependable negative dataset was essen-tial for improving the model’s prediction performance. A primary problem in miRNA–gene association studies is the scarcity of empirically validated negative data, specifically confirmed non-interactions between miRNAs and genes. To address this restriction, AEmiGAP employed an elaborate distance-based filtering technique to produce negative data, guaranteeing enhanced quality and precision compared to traditional random pairing methods. The term “distance” denotes the vector space distance between miRNAs and genes. This distance was computed using values vectorized by the Protein2Vec technique. This work developed the negative dataset based on the assumption that unknown miRNA–gene pairs with distances over the average distance of all positive pairs could be considered as negative candidates. Through this methodology, AEmiGAP guarantees that the negative set is more precise and representative than traditional random pairing techniques.

To enhance the negative dataset, we employed three different distance metrics—the Euclidean distance, cosine similarity, and Mahalanobis distance—to more accurately represent the dissimilarities between miRNA–gene pairings. This method guaranteed that the produced negative couples were clearly differentiated from the positive interactions rather than being chosen at random.

The distances for each miRNA–gene pair in the positive dataset were computed using the three metrics. Subsequently, we determined thresholds based on the average distance values from the positive dataset for each metric. Pairs that fell under these levels, showing excessive similarity to the positive interactions, were eliminated. This guaranteed that the negative dataset comprised miRNA–gene combinations that were far enough away from any established positive interaction, hence improving the quality of the negative data.

Upon applying the filtering criteria, we produced an extensive collection of negative candidate pairs. We randomly selected an equal quantity of negative pairs from this pool to equilibrate the positive dataset. A meticulously selected negative collection of 358,864 pairs, equivalent to the number of positive pairs (358,864), was created. This equilibrium is crucial in binary classification tasks, as it averts model bias towards the majority class and improves its capacity to reliably distinguish between true and false interactions.

This rigorous method of generating negative data greatly enhanced AEmiGAP’s ability to manage intricate biological datasets and elevated its overall prediction precision. By utilizing three complementing distance metrics, we verified that the negative dataset was both unique from the positive set and biologically plausible, enhancing the model’s ability to accurately identify actual miRNA–gene relationships while reducing false positives and negatives. This sophisticated methodology distinguishes AEmiGAP from models that depend exclusively on random negative pairings, offering a more dependable basis for miRNA–gene interaction studies.

### 4.4. Deep Learning Model

To identify the optimal deep learning model for predicting miRNA–gene associations, we conducted an experiment evaluating various deep learning models. For this evaluation, we used our 717,728 miRNA–gene pairs. This performance evaluation experiment did not utilize 5-fold cross-validation, as its primary aim was to determine the optimal model. Instead, we randomly divided the dataset into an 80:20 split for training and testing. The dataset configuration included the original data augmented with latent vectors produced by the autoencoder, which is the approach proposed in this study. For this evaluation, we compared the performance of four models known to excel in time series data: LSTM, Bi-LSTM, GRU, and Transformer. The Transformer model is designed with a structure that converts the input sequence to 128 dimensions and applies an eight-head multi-head attention mechanism to learn relationships between sequence elements, followed by two 128-dimensional feed-forward layers, layer normalization, and residual connections for stability and enhanced feature representation. All model evaluation experiments were conducted under the same environment and conditions, with a consistent batch size of 128 and 5,000,000 iterations, and the results are presented in Table 13.

Table 13 indicates that the LSTM model exhibited superior performance in forecasting miRNA–gene correlations relative to the other models. The improved performance is due to the LSTM’s capacity to accurately capture sequential dependencies, rendering it ideal for time-series data. In contrast to the Transformer model, which utilizes an attention mechanism, the LSTM architecture facilitates superior retention of significant information across extended sequences, proving advantageous in miRNA–gene association tasks. Consequently, we selected the LSTM model as the deep learning framework for this study.

This study employed an LSTM-based model. LSTM, a specialized kind of recurrent neural networks (RNNs), excels at handling sequential input and is especially effective in mitigating the vanishing gradient problem inherent in traditional RNNs. An LSTM unit comprises three fundamental gates:Forget Gate: This determines which information to discard from the cell state.Input Gate: This updates the cell state with new information.Output Gate: This produces the output for the current time step based on the cell state.

In this study, we employed three LSTM layers to capture long-range correlations within the miRNA and gene sequences. The quantity of hidden units in each LSTM layer was established at 128, affording an adequate capability for the acquisition of complex patterns in the data. The model was trained using a batch size of 128 with 5,000,000 iterations, allowing for comprehensive learning across the dataset.

The binary cross-entropy loss function was utilized to evaluate errors by comparing the predicted probabilities with the actual class labels. We employed the Adam optimizer, which dynamically adjusts the learning rate and efficiently updates model parameters throughout training. We utilized the Robust Scaler for data normalization to mitigate the influence of outliers, which are prevalent in biological datasets. In the output layer, the sigmoid activation function was used to produce probability scores for the miRNA–gene association predictions, outputting values between 0 and 1 for classification purposes. The integration of LSTM layers, refined parameters, and pre-processing techniques enabled the model to accurately predict miRNA–gene relationships by analyzing the sequence data.

Our overall deep learning model structure is as follows:Input Layer: The model’s input comprises the integrated miRNA and gene sequence embeddings produced by Protein2Vec, in addition to latent characteristics derived from the autoencoder. The autoencoder compresses the input data into a latent space, encapsulating vital features, which are then concatenated with the original embeddings to enrich the model’s input.LSTM Layers: The model comprises three LSTM layers that analyze the sequential data, preserving long-range dependencies throughout time steps.Fully Connected Layer: After the LSTM layers, a fully connected (dense) layer maps the LSTM output to a binary classification prediction.Output Layer: The output layer employs a sigmoid activation function, providing a probability score that indicates the likelihood of an miRNA–gene association.

## 5. Conclusions

This study marks significant progress in predicting complex genomic associations, particularly miRNA–gene interactions, through the integration of autoencoders and deep learning models. By implementing an optimized feature extraction strategy with autoencoders, AEmiGAP surpasses existing methods for miRNA–gene association prediction, achieving superior accuracy across all performance metrics. With its outstanding predictive accuracy, AEmiGAP establishes itself as a leading model, setting a new standard in the field.

AEmiGAP’s performance was rigorously validated through multiple case studies, focusing on previously unexplored miRNA–gene associations. This study identified novel candidate targets for biological validation, presenting the top 30 predicted positive associations. Specifically, we emphasized the identification of novel miRNA interactions with five major oncogenes, uncovering miRNAs potentially integral to cancer progression. These findings highlight AEmiGAP’s potential impact on cancer research, as accurate miRNA–gene association predictions can guide early diagnosis, personalized treatments, and the development of targeted therapies. Furthermore, the predicted associations provide valuable leads for laboratory validation, enabling researchers to explore new miRNA–gene interactions experimentally. These efforts can contribute to biomarker discovery and advance the understanding of cancer biology and gene regulation.

The results of this research extend beyond miRNA–gene association prediction, offering valuable insights that may inform drug discovery and deepen the understanding of complex disease mechanisms. By exploring miRNA–gene interactions with a comprehensive deep learning and feature extraction approach, AEmiGAP establishes a new benchmark for future bioinformatics research and advances in precision medicine.

Despite AEmiGAP’s remarkable performance, some limitations remain. Specifically, the lack of experimentally validated negative data presents a challenge for model validation. Future research should aim to gather empirical evidence for expected correlations or investigate more advanced methods for generating negative data to enhance model validation.

In this study, Protein2Vec was utilized to embed miRNA and gene sequences into vector representations. While effective in capturing essential sequence features, future research could explore advanced embedding techniques, such as ProteinT5 and ESM3, to capture more complex patterns and improve predictive accuracy. These protein language models are specifically designed to capture complex sequence patterns and contextual information, which could enhance the quality of embeddings and further improve the accuracy of miRNA–gene association predictions. Incorporating such state-of-the-art techniques into AEmiGAP may provide deeper insights into miRNA–gene interactions and elevate the model’s performance in future studies.

Moreover, although autoencoders demonstrated efficacy in this work, alternative unsupervised or semi-supervised learning techniques may be explored to further refine feature extraction. In particular, exploring a variety of advanced feature extraction models could provide new opportunities to capture complex patterns in biological data and enhance the quality of representations. Such efforts may uncover additional insights and further elevate the predictive performance of AEmiGAP.

AEmiGAP, initially designed for miRNA–gene relationships, could be adapted in future research to incorporate more intricate genomic interactions, including protein–protein or gene–protein interactions. Furthermore, integrating multi-omics data may yield a more comprehensive view of genomic regulation, thereby enhancing the understanding of miRNA functions and disease pathways. These expansions would further increase AEmiGAP’s applicability and utility in precision medicine.

In summary, while AEmiGAP establishes a novel benchmark for miRNA–gene association prediction, future efforts will seek to address these limitations and expand its applicability to other genomic research domains. These advancements will further the fields of precision medicine and therapeutic innovation, ensuring continued progress in understanding and targeting complex biological systems.

## Figures and Tables

**Figure 1 ijms-25-13075-f001:**
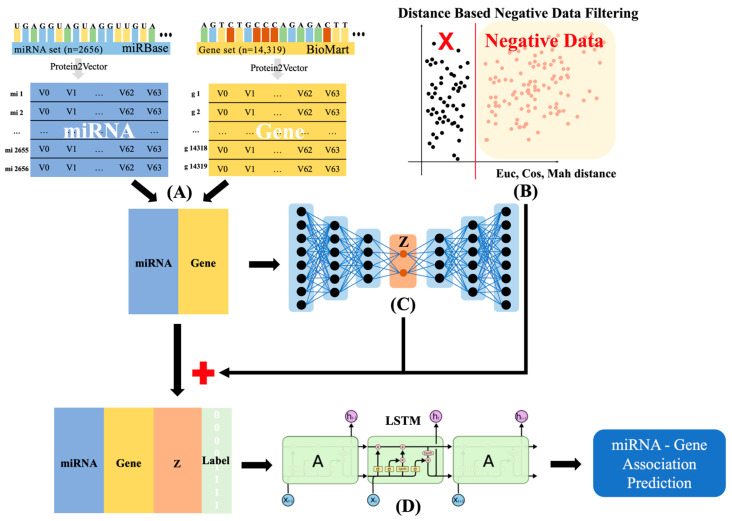
The overall workflow of the AEmiGAP model for miRNA–gene association prediction. (**A**) Protein2Vector embeddings of miRNA and gene sequence data from miRBase and BioMart. (**B**) Negative data filtering based on Euclidean, cosine, and Mahalanobis distances. (**C**) Autoencoder to extract latent vector features from the miRNA–gene pairs. (**D**) LSTM-based deep learning model for final miRNA–gene association prediction.

**Figure 2 ijms-25-13075-f002:**
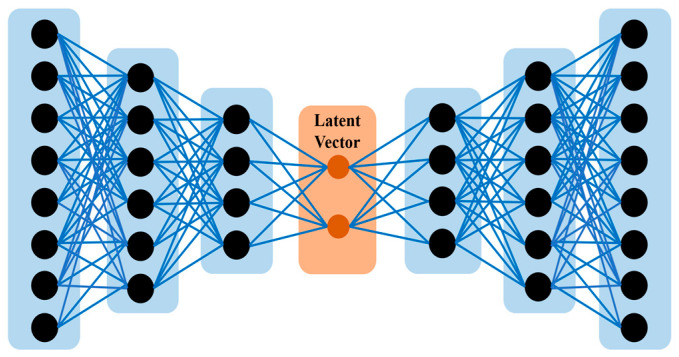
The structure of the autoencoder model used in our study. The encoder compresses input data into a latent vector, and the decoder reconstructs the original data from this compressed form. This model is designed to capture complex features from miRNA and gene sequences, improving the deep learning model’s performance.

**Figure 3 ijms-25-13075-f003:**
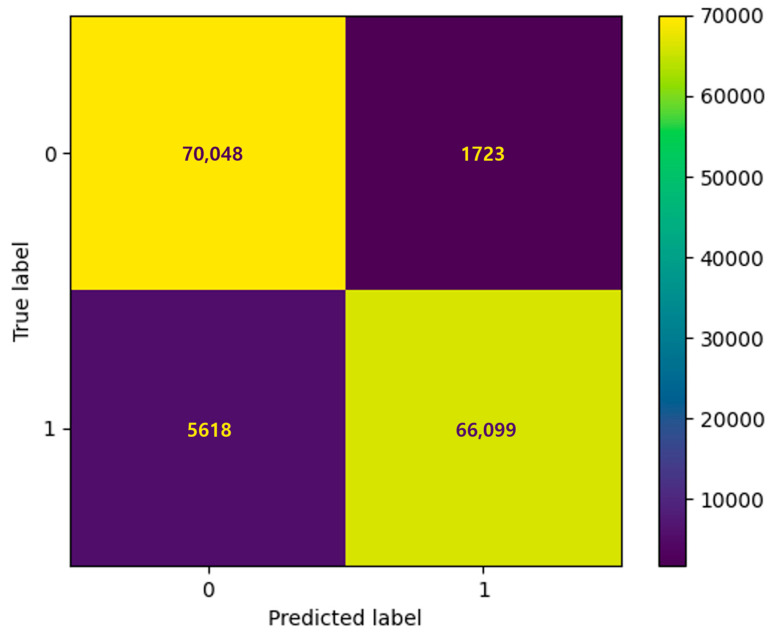
Receiver Operating Characteristic (ROC) curve and confusion matrix for Fold-4, the best-performing fold in the 5-fold cross-validation of the AEmiGAP model. The ROC curve demonstrates a high Area Under the Curve (AUC) value of 0.9857, indicating excellent model performance in distinguishing between positive and negative miRNA–gene associations. The confusion matrix shows the distribution of true positives, false positives, true negatives, and false negatives. Fold-4 achieves the best balance between precision and recall, resulting in the highest overall accuracy and F1 score among the five folds.

**Table 1 ijms-25-13075-t001:** Overview of recent miRNA–genome research studies. This table summarizes key research studies that explore the relationships between miRNAs and various genomic elements. It lists each study’s association objectives, utilized models, data sources, and computational approaches. The positive and negative dataset sizes, embedding techniques, and main results are also provided to give a comprehensive view of the methodologies and outcomes in recent miRNA–genome interaction research.

miRNA Association Object	Model Name	Dataset					Negative Data Method	Embedding Method	Computational Model	Results
		miRNA Sequence Data	Object Sequence Data	miRNA–Object Relation Data	Positive	Negative				
Gene	SG-LSTM FRAME [6] (2021)	miRbase [7]	biomaRT [8]	miRTarBase [9]	15,540	15,540	Euclidean [10], Cos [11], Sim	Doc2Vec [12], Role2Vec [13]	LSTM [14]	AUC 0.9363
	miTAR [15] (2021)	miRbase	-	DeepMirTar [16], miRAW [17]	3908	3898	MiRanda	Embedding Layer	CNN [18], Bi-RNN [19]	Accuracy 95.5
	MDCNN [20] (2021)	miRbase	HumanNet [21]	mirTarBase	237,574	-	Random Sampling	One-Hot-Encoding	HIN, DCNN	AUC 0.9096
	SRG-Vote [22] (2022)	miRbase	biomaRT	SG-LSTM-whole	15,540	15,540	Euclidean, Cos, Sim	Doc2Vec, Role2Vec, GCN [23]	LSTM, Bi-LSTM [24]	AUC 0.95
	miGAP (2023)	miRbase	biomaRT	mirTarBase	358,864	358,864	Euclidean, Cos, Sim, Mahalanobis	Proten2Vec	LSTM	AUC 0.9834
mRNA	MiRTDL [25] (2015)	miRbase	NCBI [26]	TarBase [27]	1297	309	-	-	CNN	Accuracy 0.96
	deepTarget [28] (2016)	miRbase	-	miRecords [29]	3398	3640	Fisher–Yates shuffle, MiRanda	Dense vector	Autoencoder	Accuracy 0.9641
	DeepMirTar (2018)	miRbase	UCSC Genome Browser [30]	mirMark [31], CLASH [32]	3915	3905	MiRanda	One-Hot-Encoding	AE	AUC 0.9793
	IMTRBM [33] (2018)	miRbase	NCBI	miRTarBase, BioGRID [34]	8420	-	-	-	RBM	-
	miRAW [17] (2018)	miRBase	Diana TarBase, MirTarBase	Diana TarBase, miRTarBase	303,912	32,284	RNACoFold [35]	One-Hot-Encoding	AE, FFN	AUC 0.96
	miTDS [36] (2024)	miRAW	miRAW	miRAW	548	548	-	BERT	Attention	Accuracy 0.78
lncRNA	LncMirNet [37] (2020)	miRbase	GENCODE [38]	lncRNASNP2 [39]	15,386	15,386	Knuth–Durstenfeld	K-mer, CTD, Doc2Vec, Role2Vec	CNN	AUC 0.9381
	TEC-LncMir [40] (2024)	miRbase	GENCODE	lncRNASNP2	15,386	15,386	Knuth–Durstenfeld shuffle algorithm	K-mer, Transformer Encoder	Transformer, CNN	AUC 0.9544
circRNA	WSCD [41] (2022)	CircBank [42], miRBase	CircBank	CircBank	20,208	20,208	Random Sampling	Wored2vec, SDNE [43]	CNN, DNN	AUC 0.8988
	NGCICM [44] (2023)	CircR2Cancer [45]	circBase [46]	CircR2Cancer	753	753	Random Sampling	Node2vec [47]	GAT	AUC 0.9697 (2-fold), 0.9932 (5-fold)
	DeepCMI [48] (2024)	CircBank, CircR2Cancer	CircBank, CircR2Cancer	CircBank, CircR2Cancer	9905	9905	Random Sampling	Multi-Source Feature, Gaussian Interaction Profile	DeepWalk [49], XGBoost [50]	AUC 90.48

**Table 2 ijms-25-13075-t002:** Performance comparison of miRNA–gene association prediction using different latent vector sizes. The table presents the AUC scores for two experiments: (A) using only the latent vectors produced by the autoencoder, and (B) combining the latent vectors with the original sequence data from Protein2Vec.

Latent Vector Size	Latent Vector (A)	Original + Latent Vector (B)
8	0.9022	0.9818
16	0.8988	0.9837
32	0.9229	0.9832
64	0.9231	**0.9846**
128	0.9331	0.9840

**Table 3 ijms-25-13075-t003:** Five-fold cross-validation prediction performance results, showing precision, recall, accuracy, F1 score, and AUC across five folds. The average results are summarized at the bottom, highlighting the high overall performance of the AEmiGAP model across various evaluation metrics.

	Precision	Recall	Accuracy	F1 Score	AUC
Fold-1	0.966	0.923	0.943	0.945	0.9814
Fold-2	0.944	0.941	0.942	0.942	0.9854
Fold-3	0.961	0.931	0.945	0.946	0.9852
Fold-4	0.976	0.926	0.949	0.950	**0.9857**
Fold-5	0.982	0.921	0.949	0.950	0.9851
Average	0.9658	0.9284	0.9456	0.9466	0.9846

**Table 4 ijms-25-13075-t004:** Dataset description for the case study aiming to discover new miRNA–gene associations. The table includes the total number of miRNA and gene entities in the dataset, which comprises 2656 miRNAs and 14,319 genes, resulting in 37,672,400 possible miRNA–gene pairs. All miRNA–gene pairs documented in miRTarBase were excluded to ensure the discovery of novel associations. This curated Unknown Data Pool was utilized to explore and identify miRNA–gene connections that remain unconfirmed in existing databases.

Data	Data Count
Unknown Data Pool	37,672,400
(miRNA)	2656
(Gene)	14,319

**Table 5 ijms-25-13075-t005:** The top 30 miRNA–gene pairs predicted by the AEmiGAP model from the Unknown Data Pool were ranked according to their predicted correlation scores. These pairs are not present in the miRTarBase database but have been evaluated using TargetScan and miRWalk for potential biological significance. The table highlights miRNA and gene pairings that suggest possible novel associations, which could be valuable for further biological validation.

Rank	miRNA	Gene	Support
1	hsa-miR-92a-3p	*C1D*	
2	hsa-miR-523-5p	*TMEM126B*	TargetScan
3	hsa-miR-548ah-5p	*IKBIP*	TargetScan
4	hsa-miR-4728-5p	*ZNF267*	
5	hsa-miR-548ah-5p	*PNISR*	
6	hsa-miR-92a-3p	*PTPN12*	
7	hsa-miR-30c-5p	*FYB2*	
8	hsa-miR-92a-3p	*PTBP3*	miRwalk
9	hsa-miR-92a-3p	*SCOC*	miRwalk
10	hsa-miR-4485-3p	*ASPSCR1*	
11	hsa-miR-6794-3p	*GDE1*	TargetScan
12	hsa-miR-92a-3p	*RBL2*	TargetScan
13	hsa-miR-92a-3p	*LRRFIP1*	miRwalk
14	hsa-miR-92a-3p	*IVL*	
15	hsa-miR-548ah-5p	*LRRFIP1*	
16	hsa-miR-4485-3p	*RTBDN*	
17	hsa-miR-4707-5p	*PITX3*	miRwalk
18	hsa-miR-92a-3p	*BIVM*	miRwalk
19	hsa-miR-548ah-5p	*MED13L*	
20	hsa-miR-92a-3p	*SRP9*	
21	hsa-miR-92a-3p	*NBR1*	
22	hsa-miR-548ah-5p	*USP28*	
23	hsa-miR-4485-3p	*DPP8*	
24	hsa-miR-548ah-5p	*BORCS7*	
25	hsa-miR-92a-3p	*TAF3*	miRwalk
26	hsa-miR-92a-3p	*MSH3*	
27	hsa-miR-92a-3p	*GNAI1*	miRwalk
28	hsa-miR-92a-3p	*C9orf78*	
29	hsa-miR-548ah-5p	*FBXO34*	
30	hsa-miR-519c-5p	*TMEM126B*	TargetScan

**Table 6 ijms-25-13075-t006:** Top 10 candidate miRNAs associated with ALK from the Unknown Data Pool.

Rank	miRNA	Gene	Support
1	hsa-miR-6081	*ALK*	
2	hsa-miR-4708-3p	*ALK*	miRWalk
3	hsa-miR-1225-3p	*ALK*	
4	hsa-miR-6808-5p	*ALK*	
5	hsa-miR-4707-3p	*ALK*	
6	hsa-miR-6836-5p	*ALK*	
7	hsa-miR-6075	*ALK*	
8	hsa-miR-744-5p	*ALK*	
9	hsa-miR-494-3p	*ALK*	
10	hsa-miR-6720-5p	*ALK*	

**Table 7 ijms-25-13075-t007:** Top 10 candidate miRNAs associated with GNAS from the Unknown Data Pool.

Rank	miRNA	Gene	Support
1	hsa-miR-519c-5p	*GNAS*	miRWalk
2	hsa-miR-4485-3p	*GNAS*	
3	hsa-miR-518d-5p	*GNAS*	
4	hsa-miR-6894-5p	*GNAS*	miRWalk
5	hsa-miR-523-5p	*GNAS*	miRWalk
6	hsa-miR-4512	*GNAS*	miRWalk
7	hsa-miR-518f-5p	*GNAS*	
8	hsa-miR-548a-5p	*GNAS*	
9	hsa-miR-6129	*GNAS*	
10	hsa-miR-522-5p	*GNAS*	miRWalk

**Table 8 ijms-25-13075-t008:** Top 10 candidate miRNAs associated with KRAS from the Unknown Data Pool.

Rank	miRNA	Gene	Support
1	hsa-miR-25-3p	*KRAS*	miRWalk
2	hsa-miR-4753-3p	*KRAS*	miRWalk, TargetScan
3	hsa-miR-6826-3p	*KRAS*	miRWalk, TargetScan
4	hsa-miR-4685-3p	*KRAS*	
5	hsa-miR-204-5p	*KRAS*	
6	hsa-miR-18b-5p	*KRAS*	miRWalk
7	hsa-miR-6894-3p	*KRAS*	TargetScan
8	hsa-miR-6165	*KRAS*	miRWalk
9	hsa-miR-615-3p	*KRAS*	miRWalk
10	hsa-miR-3681-5p	*KRAS*	miRWalk

**Table 9 ijms-25-13075-t009:** Top 10 candidate miRNAs associated with EGFR from the Unknown Data Pool.

Rank	miRNA	Gene	Support
1	hsa-miR-25-3p	*EGFR*	miRWalk
2	hsa-miR-4685-3p	*EGFR*	miRWalk, TargetScan
3	hsa-miR-6794-3p	*EGFR*	
4	hsa-miR-18b-5p	*EGFR*	miRWalk
5	hsa-miR-6894-3p	*EGFR*	TargetScan
6	hsa-miR-6829-3p	*EGFR*	miRWalk, TargetScan
7	hsa-miR-4753-3p	*EGFR*	miRWalk, TargetScan
8	hsa-miR-6780a-3p	*EGFR*	miRWalk, TargetScan
9	hsa-miR-6754-3p	*EGFR*	miRWalk, TargetScan
10	hsa-miR-6081	*EGFR*	TargetScan

**Table 10 ijms-25-13075-t010:** Top 10 candidate miRNAs associated with PIK3CA from the Unknown Data Pool.

Rank	miRNA	Gene	Support
1	hsa-miR-25-3p	*PIK3CA*	TargetScan
2	hsa-miR-6826-3p	*PIK3CA*	miRWalk, TargetScan
3	hsa-miR-6780a-3p	*PIK3CA*	miRWalk
4	hsa-miR-4753-3p	*PIK3CA*	TargetScan
5	hsa-miR-6829-3p	*PIK3CA*	TargetScan
6	hsa-miR-204-5p	*PIK3CA*	miRWalk
7	hsa-miR-3681-5p	*PIK3CA*	miRWalk
8	hsa-miR-18b-5p	*PIK3CA*	miRWalk
9	hsa-miR-4707-3p	*PIK3CA*	
10	hsa-miR-6081	*PIK3CA*	

**Table 11 ijms-25-13075-t011:** Performance comparison between AEmiGAP and miGAP models across key evaluation metrics (average performance over 5-fold cross-validation). The table highlights improvements in precision, recall, accuracy, F1 score, and AUC, demonstrating the enhanced predictive capability of AEmiGAP over miGAP.

(AVG)	Precision	Recall	Accuracy	F1 Score	AUC
miGAP	0.938	0.9278	0.934	0.9326	0.9804
**AEmiGAP**	**0.** **9658**	**0.** **9284**	**0.** **9456**	**0.** **9466**	**0.** **9846**
Improvement	+2.97%	+0.06%	+1.24%	+1.50%	+0.43%

**Table 12 ijms-25-13075-t012:** Comparison of recent miRNA–gene association studies with AEmiGAP.

miRNA Association Object	Model Name	Dataset					Negative Data Method	Embedding Method	Computational Model	Results (Best)
		miRNA Sequence Data	Object Sequence Data	miRNA–Object Relation Data	Positive	Negative				
Gene	SG-LSTM FRAME (2021)	miRbase	biomaRT	miRTarBase	15,540	15,540	Euclidean, CosSim	Doc2Vec, Role2Vec	LSTM	AUC 0.9363
	miTAR (2021)	miRbase	-	DeepMirTar, miRAW	3908	3898	MiRanda	Embedding Layer	CNN, Bi-RNN	Accuracy 95.5
	MDCNN (2021)	miRbase	HumanNet	miRTarBase	237,574	-	Random Sampling	One-Hot-Encoding	HIN, DCNN	AUC 0.9096
	SRG-Vote (2022)	miRbase	biomaRT	miRTarBase	15,540	15,540	Euclidean, CosSim	Doc2Vec, Role2Vec, GCN	LSTM, Bi-LSTM	AUC 0.95
	**AEmiGAP** **(2024)**	**miRbase**	**biomaRT**	**miRTarBase**	**358,864**	**358,864**	**Euclidean, CosSim, Mahalanobis**	**Proten2Vec, autoencoder**	**LSTM**	**AUC** **0.9857**

**Table 13 ijms-25-13075-t013:** Performance comparison of different deep learning models for miRNA–gene association prediction, showing the AUC scores for each model. Among the tested models, the LSTM demonstrated the highest AUC score, indicating superior performance in capturing miRNA–gene associations, followed by GRU, Bi-LSTM, and Transformer.

	LSTM	Bi-LSTM	GRU	Transformer
AUC	**0.9846**	0.9638	0.9478	0.9436

## Data Availability

The original contributions presented in this study are included in the article. Further inquiries can be directed to the corresponding author.

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
