# Peer review of "AEmiGAP: AutoEncoder-Based miRNA–Gene Association Prediction Using Deep Learning Method"

_ijms, 2024, doi:10.3390/ijms252313075_

Round 1
Reviewer 1 Report (New Reviewer)
Comments and Suggestions for Authors
The authors present AEmiGAP, a deep learning model that integrates autoencoders with LSTM networks for miRNA-gene association prediction. By leveraging advanced feature extraction methods, AEmiGAP enhances the accuracy of identifying miRNA-gene relationships, which is significant for cancer genomics and precision medicine. The paper reports promising results in predictive accuracy, outperforming existing models and providing new insights through case studies. Before their works to be accepted to publish in our journal, some revisions are need to make.
Major comments:
1. Is the gene sequence used in the study from the cancer genomics? Please describe the source of the gene sequences in detail.
2. The study mentions the use of distance metrics to generate negative samples. Please provide more details to assess the validity and reliability of this method.
3. The authors mentioned using protein2Vec to convert protein sequences into vectors in this paper, the authors can try protein language models such as ProteinT5 or ESM3.
4. Did the authors consider the possibility of data leakage when evaluating the method, which could lead to an overestimation of the efficacy of AEmiGAP. For example, how the dataset was divided and whether sequences in the test set were included in the training set.
5. The use of autoencoders for feature extraction is central to the model, but more evidence is needed to show why this approach outperforms other feature extraction techniques. Have the authors considered evaluating the performance of other models for feature extraction, such as Pretrained Models.
6. The model section of the paper mentions the use of specific parameter settings, such as learning rate, batch size, and so on. Could the authors please describe in detail the rationale for choosing these parameters, and whether parameter optimization was performed.
7. While AEmiGAP performs well on the curated dataset, it lacks evaluation on independent external benchmarks to validate its generalizability across diverse biological contexts.
8. Why choose LSTM instead of other networks as the backbone of the model, more ablation experiments are needed to prove its superiority.
Author Response
Please see the attachment.

Reviewer 2 Report (New Reviewer)
Comments and Suggestions for Authors
In the present work Yoon et al. presented a new algorithmic approach for the detection of miRNA/RNA interactions, using deep learning methodologies.
The overall concept is interesting, yet I am not sure that this manuscript is compatible with a journal that publishes on molecular sciences (since this is a purely bioinformatics work). Yet, apart from that remark, the present work is well-written and presents an interesting topic. RNA molecules interactions prediction is crucial for the understanding of gene regulatory dynamics. I only have a few minor comments for the authors. First of all, the authors should clearly state the scope of their work at the end of the “Introduction” section. Do they intended to present an algorithm, an integrated approach, an application that would perform their approach available to other fellow scientists? This should be clarified.
Further on, the authors have stated how their approach compares to other, previously described, methodologies. They should also highlight their findings and report on how their approach can support research. How is their approach better, not only in terms of performance, but also in terms of biological significance? How is their work going to help laboratory research for the prediction and further investigation of target molecules?
Round 2
Reviewer 1 Report (New Reviewer)
Comments and Suggestions for Authors
The comments 4-6 in the file of “author_response.pdf” are the same and wrongly typed, please revise.
Author Response
Thank you for bringing to our attention the issue in the "author_response.pdf” file.
We sincerely apologize for this oversight and have thoroughly reviewed and corrected the document to ensure all comments are appropriately addressed.
We have ensured that the responses to reviewer comments are now accurate and properly aligned with the feedback received.
Thank you for your continued support and consideration.
Best regards,
Seungwon Yoon

This manuscript is a resubmission of an earlier submission. The following is a list of the peer review reports and author responses from that submission.
Round 1
Reviewer 1 Report
Comments and Suggestions for Authors
The structure and content of the article need further improvement and detailed explanation to enhance its readability and scientific validity. There are some problems should be addressed to improve the current manuscript.
1. The abstract section is written in a sloppy way, the authors do not introduce clearly what is missing in the field and how their methods aim at improving upon existing ones.
2. Please clearly present the major innovation of this work.
3. Although the paper mentions using 5-fold cross-validation for model evaluation, it does not provide a detailed discussion of the model's generalization capability and performance across different datasets. It is recommended that the authors further validate the model's performance on different, independent test sets and discuss the model's stability and generalization ability across various datasets.
4. The authors say that the AEmiGAP model outperforms existing miRNA-gene association prediction methods but do not provide direct comparisons with existing models. It is recommended that the authors provide a detailed comparison of AEmiGAP with other state-of-the-art models.
5. The discussion section should more comprehensively address the limitations of the model, potential areas for improvement, and the potential impact of the model in practical applications.
Comments on the Quality of English LanguageThe English is understandable.
Reviewer 2 Report
Comments and Suggestions for Authors
MicroRNAs constitute a class of endogenous, small non-coding RNAs of approximately 20-24 nucleotides in length, which perform a variety of crucial regulatory functions within cells. Each microRNA has the potential to regulate multiple target genes, and several miRNAs may also regulate the same gene. In this manuscript, autoencoders are employed to extract features for the characterization of intricate, latent relationships between microRNAs and genes. A long short-term memory network is used to construct a model for predicting microRNA-gene associations. A series of experiments have demonstrated the efficacy of the current method. However, several issues remain to be addressed.
1. Euclidean, cosine and Mahalanobis distances are used to construct the negative microRNA-gene association pairs. However, it is unclear from which distance method the results of the current study are derived. The authors do not provide sufficient detail on this point.
2. It can be observed that the data and content present in Table 13 are replicated in Table 1. Furthermore, the comparison is inherently flawed and inconsequential given the disparate methodologies and data sets employed.
3. It is well known that the number of negative samples is considerably greater than that of positive samples. However, the current study has only constructed the model based on a ratio of positive to negative samples of 1:1. It would be of interest to ascertain whether the current method would still yield excellent results if this ratio is changed to 1:10.
Reviewer 3 Report
Comments and Suggestions for Authors
While the topic is relevant and timely, there are a few issues need to be addressed to enhance the clarity of the work.
- The current figures' resolution is rather low. Please fix this issue.
- The manuscript seems to include too many short sections. It is better to simplify the structure by merging some of the sections that ideas flow subject-wise.
- The paper primarily provides descriptions of various imputation techniques without delving deeply into their comparative advantages, disadvantages, or suitability for specific scenarios.
- I recommend that the authors include a comprehensive review of relevant articles in the literature section. i.e. https://doi.org/10.3390/pharmaceutics15082061, https://doi.org/10.1007/s00521-024-09518-z
- It would be more appropriate to use passive voice and avoid personal pronouns (we, you, your) to maintain a more objective scientific language of the paper.
- The authors should draw a flowchart for the proposed model to be more clear for the readers? Create a flowchart that visually represents the steps and components of the proposed model. Also, it Include details such as data preprocessing, feature extraction, model training, and prediction.
Reviewer 4 Report
Comments and Suggestions for Authors
AEmiGAP is a deep learning model combining autoencoders with LSTM networks to predict miRNA-gene interactions, focusing on improved feature extraction. It creates a dataset of positive and negative miRNA-gene pairs. The model demonstrated higher accuracy and AUC compared to and improved previous methods. The authors list potential miRNA-gene interactions related to oncogene genes.
Major comments:
1. Introduction: Rewrite the introduction in a concise way.
2. Overall Organization: The manuscript will benefit from being rewritten. No need for 13 tables can be reduced to 4-5 informative tables. Same applied for figures that could be reduced to two informative ones.
3. The presented method is an extension of miGAP (as mentioned and published by the same author); this fact must be mentioned clearly in the abstract to better understand the novelty. AEmiGAP is based on the original papers and on preprocessing from the miGAP project (miGAP: miRNA-Gene Association Prediction Method Based on Deep Learning Model, Appl. Sci. 2023). Clearly mention the difference to miGAP. Also, when comparing methods, this should be one of the key methods to compare to.
4. Table 1 is relevant. A detailed summary in words is not needed (unless you want to highlight something specific). Remove all the dense 1.5 pages that repeat the content of Table 1 and its description.
5. comparisons: When comparing the different tools, including (e.g., NGCICM, DeepCMI, WSCD models), the authors report on very high performance, with AUC being very high (0.9-0.99). So, it is not clear what ‘gap’ in knowledge they are addressing.
6. Data accessibility: For transparency and reproducibility, supplementary tables that show results for all drivers (and not selected genes and selected 10 top ones) should be accessible for further evaluations. The reader will benefit from such information that seems not available in this manuscript. Also without such information it is not easy to assess the difference on top 10 and bottom 10 miRNA-gene pairs and other assessment becomes limited.
7. It is not clear what makes the method particularly relevant to cancer (besides selecting manually several key drivers from COSMIC data). Please clarify whether there are any specific properties of the method that preferably match cancer research.
Minor comments:
1. The introduction is read as a review article on the field of miRNA-gene interaction and is too general. The readers can be forwarded to a few references and it can be reduced to 1-2 informative paragraphs.
2. Remove Table 2. The number should be mentioned in the methods section as a text.
3. Figure 2b: represent differently (as a histogram using bins or zoom in for the first 10 epochs).
4. Avoid too many general statements (section 5). Some of the sentences are too broad and add minimal concrete information and cannot be considered as new conclusions.
5. Remove Figure 3. It is not adding too much to the flow.
6. Maybe easier to read the title by adding the word “using” deep learning…